# Autophagy inhibition overcomes multiple mechanisms of resistance to BRAF inhibition in brain tumors

Jean M Mulcahy Levy[1,2]\*, Shadi Zahedi[1,2], Andrea M Griesinger[1,2], Andrew Morin[1,2], Kurtis D Davies[3], Dara L Aisner[3], BK Kleinschmidt-DeMasters[3,4], Brent E Fitzwalter[5], Megan L Goodall[5], Jacqueline Thorburn[5], Vladimir Amani[1,2], Andrew M Donson[1,2], Diane K Birks[2,4], David M Mirsky[2,6], Todd C Hankinson[2,4], Michael H Handler[2,4], Adam L Green[1,2], Rajeev Vibhakar[1,2], Nicholas K Foreman[1,2], Andrew Thorburn[5]

[1]Department of Pediatrics, University of Colorado Denver, Aurora, United States; [2]Morgan Adams Foundation Pediatric Brain Tumor Research Program, Children's Hospital Colorado, Aurora, United States; [3]Department of Pathology, University of Colorado Denver, Aurora, United States; [4]Department of Neurosurgery, University of Colorado Denver, Aurora, United States; [5]Department of Pharmacology, University of Colorado Denver, Aurora, United States; [6]Department of Radiology, University of Colorado Denver, Aurora, United States

\*For correspondence: Jean.
MulcahyLevy@ucdenver.edu

Competing interests: The authors declare that no competing interests exist.

**Abstract** Kinase inhibitors are effective cancer therapies, but tumors frequently develop resistance. Current strategies to circumvent resistance target the same or parallel pathways. We report here that targeting a completely different process, autophagy, can overcome multiple BRAF inhibitor resistance mechanisms in brain tumors. $BRAF^{V600E}$ mutations occur in many pediatric brain tumors. We previously reported that these tumors are autophagy-dependent and a patient was successfully treated with the autophagy inhibitor chloroquine after failure of the $BRAF^{V600E}$ inhibitor vemurafenib, suggesting autophagy inhibition overcame the kinase inhibitor resistance. We tested this hypothesis in vemurafenib-resistant brain tumors. Genetic and pharmacological autophagy inhibition overcame molecularly distinct resistance mechanisms, inhibited tumor cell growth, and increased cell death. Patients with resistance had favorable clinical responses when chloroquine was added to vemurafenib. This provides a fundamentally different strategy to circumvent multiple mechanisms of kinase inhibitor resistance that could be rapidly tested in clinical trials in patients with $BRAF^{V600E}$ brain tumors.

## Introduction

Signaling pathway-targeted therapies in cancer are greatly hampered by our inability to counteract the development of resistance. The RAF/MEK/ERK pathway is important in central nervous system tumors (*Gierke et al., 2016*; *Mistry et al., 2015*), and with $BRAF^{V600E}$ mutations in more than 50% of select tumors (*Penman et al., 2015*) there is great potential for the use of $BRAF^{V600E}$ inhibitors. Indeed, the first pediatric patient successfully treated with vemurafenib (*Rush et al., 2013*) was followed by similar case reports in brain tumor patients of all ages (*Bautista et al., 2014*; *Skrypek et al., 2014*), and clinical trials in children and adolescents are ongoing using both vemurafenib (NCT01748149) and dabrafenib (NCT01677741). The initial excitement for BRAF inhibitors (BRAFi) in other tumors was tempered because the majority of patients who initially respond to RAF

**eLife digest** Cancers of the brain and spine are the second most common kind of tumor in children, after cancers of the blood and bone marrow. Yet brain and spine tumors kill more children than any other cancer, in part because many become resistant to treatment.

Like in other cancers, cells in brain and spine tumors often use a process called autophagy to survive the treatments that are used to try and kill them. This process allows cells to recycle proteins and other things inside the cell and use them for energy when the cell is under stress. In 2014, researchers reported that brain tumors carrying a mutation called $BRAF^{V600E}$ rely on autophagy to survive treatment with medications that target this mutation. These findings suggested that blocking autophagy might make the medications more effective against $BRAF^{V600E}$ mutant tumors and overcome the resistance.

Now, Mulcahy Levy et al. – who include most of the researchers involved in the 2014 study – report that blocking autophagy does indeed overcome this kind of resistance in multiple types of tumor. The experiments made use of human brain tumor cells that can be grown in the laboratory and have been widely studied, as well as samples collected from patients.

Mulcahy Levy et al. were able to block autophagy in the tumor cells by using genetic methods and, importantly, by using an approved and inexpensive drug that could be rapidly translated into clinical trials. Together these findings suggest that blocking autophagy in patients might be a safe and effective strategy to improve their response to existing therapies that target the $BRAF^{V600E}$ mutation. Future clinical trials are now needed to test more patients and verify if this treatment plan can be broadly effective in patients with these types of brain cancers.

inhibition quickly develop resistance to therapy (*Hartsough et al., 2014*; *Sun et al., 2014*). This is a significant issue in brain tumors as well (*Levy et al., 2014*; *Yao et al., 2015*).

There are multiple routes of acquired resistance to RAF inhibition (*Sun et al., 2014*; *Rizos et al., 2014*) and circumventing these mechanisms usually involves either targeting the same pathway a different way or targeting a similar parallel pathway. A recent study of BRAFi resistance in colorectal cancer highlighted difficulties with this approach with a single tumor often harboring more than one mechanism of resistance. More importantly, when tumors became resistant to one combination of drugs, such as BRAF/MEK inhibition, there was cross-resistance to others such as BRAF/EGFR inhibition (*Ahronian et al., 2015*). This concept is playing out in clinical trials as well. BRAF and MEK inhibition in $BRAF^{V600E}$ melanoma patients found a small increase in median progression free survival but failed after a short time. Further evidence found that patients who were treated with MEKi after they had developed BRAFi resistance had no objective clinical responses (*Kim et al., 2013*). EGFR is another potential secondary target in melanoma, brain, and colorectal cancer. Although encouraging preclinical results have been obtained in these tumors (*Yao et al., 2015*; *Corcoran et al., 2012*; *Girotti et al., 2013*), combined BRAF/EGFR inhibition similarly leads to incomplete and short-term responses in people (*Ahronian et al., 2015*; *Pietrantonio et al., 2016*).

Autophagy inhibition is a potential method to reverse BRAFi resistance. Previous studies of kinase inhibitor resistance in adult $BRAF^{WT}$ gliomas with PTEN mutations resistant to phosphatidylinositol 3-kinase to AKT to mammalian target of rapamycin (PI3K-AKT-mTOR) pathway inhibitors found that autophagy inhibition improved response to dual PI3K-mTOR inhibitors (*Fan et al., 2010*). Up-regulation of endoplasmic reticulum (ER) stress-induced autophagy after treatment with BRAFi has been shown in melanoma tumor biopsies and associated with the development of resistance to vemurafenib. Autophagy inhibition overcame the resistance through this mechanism in melanoma cell lines (*Ma et al., 2014*). Previously, we reversed clinical and radiographic disease progression with the addition of the autophagy inhibitor chloroquine (CQ) in a patient with a $BRAF^{V600E}$ brainstem ganglioglioma who progressed while on vemurafenib (*Levy et al., 2014*). This patient continued to experience disease regression on the combination of CQ plus vemurafenib for more than two and a half years, contrasting dramatically with her original response to vemurafenib that failed at 11 months (*Levy et al., 2014*). These findings led us to hypothesize that autophagy inhibition provides a different way to circumvent BRAF inhibitor resistance in CNS tumors that avoids targeting the

same or similar kinase pathways and might apply to multiple different mechanisms of kinase inhibitor resistance.

## Results

### Pharmacologic inhibition of autophagy overcomes BRAFi resistance in vitro

Isogenic BRAFi resistant brain tumor cell lines (794R and AM38R) were developed through chronic exposure to vemurafenib (*Figure 1A* and quantification *Figure 1B*). Parental cells (794 and AM38) demonstrated a stable reduction the in ratio of pERK to ERK when treated with vemurafenib. In contrast, resistant cells recovered pERK:ERK ratios to almost baseline levels by 24 hr of drug exposure (*Figure 1C* and quantification of pERK:ERK ratios *Figure 1D*). Unlike reports in vemurafenib-resistant melanoma cell lines (*Ma et al., 2014*), neither basal nor drug-induced autophagy was increased in resistant cells (794R, AM38R, B76) compared to parental/sensitive cells (794, AM38, BT40) as determined by flow cytometry (*Figure 2A and B*) or by Western blot (*Figure 2C*). Quantification of autophagic flux measured by Western blot (*Figure 2D*) demonstrated that 794 and 794R cells have a similar flux at all timepoints, while AM38R had a smaller accumulation of LC3II over six hours compared to AM38 parental cells. This may be in part to the AM38 parental cells that demonstrated a lower level of LC3II at baseline compared to AM38R cells. As flux is measured by comparison of all time-points to the time 0 baseline, the higher level of LC3II in AM38R cells at time 0 would reduce the final flux measurement at six hours. Taken together, these data demonstrated that development of resistance to BRAFi did not increase levels of autophagy in these cells. Additionally, autophagic flux was successfully blocked in both parental and resistant cells using 5 µM CQ (*Figure 2E*), a dose that can be achieved clinically (*Augustijns et al., 1992*).

Resistant cells retained a dose dependent sensitivity to pharmacologic autophagy inhibition equal to their parental controls in long-term growth assays and approximately 50% growth inhibition was achieved using 5 µM CQ (*Figure 3A*). Long-term growth assays demonstrated that the parental cells responded to both vemurafenib and CQ alone, and the combination resulted in an even greater reduction in cell growth as we previously reported (*Levy et al., 2014*). In comparison, resistant cells

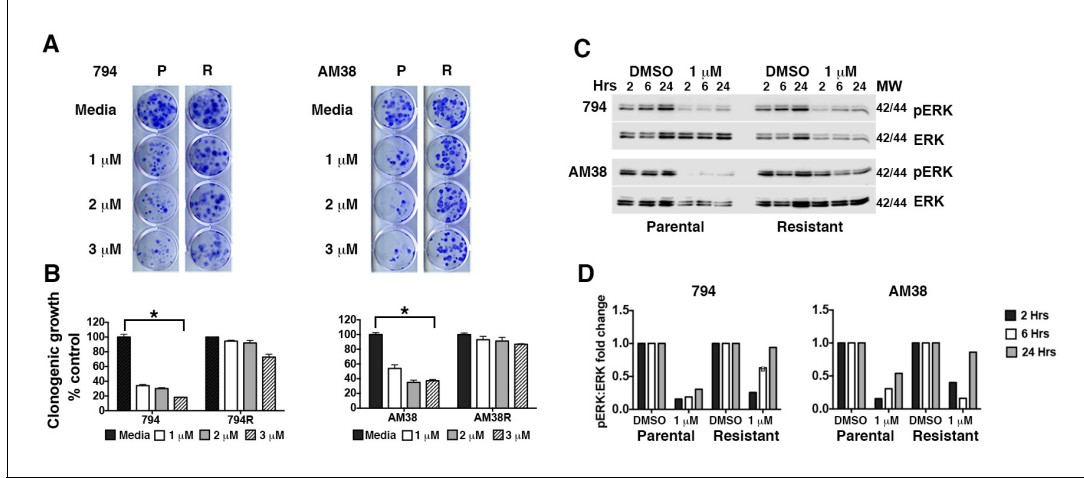

**Figure 1.** Brain tumor cell lines develop resistance to pharmacologic inhibition of BRAF$^{V600E}$. (**A**) Comparison of parental (**P**) and isogenic resistant (**R**) cell line response long-term growth following BRAFi for 72 hr. Representative image shown. (**B**) Quantification of clonogenic growth shown in **A**. Two way ANOVA; mean ± s.e.m, n = 3. *p<0.05. (**C**) Representative Western blot demonstrating decreased pERK suppression in resistant cells compared to parental cells following BRAFi. (**D**) Quantification of pERK:ERK ratios shown in **C**.

The following source data is available for figure 1:

**Source data 1.** Quantification of long-term clonogenic growth assays in 794 and AM38 parental and resistant cells treated with increasing doses of vemurafenib.

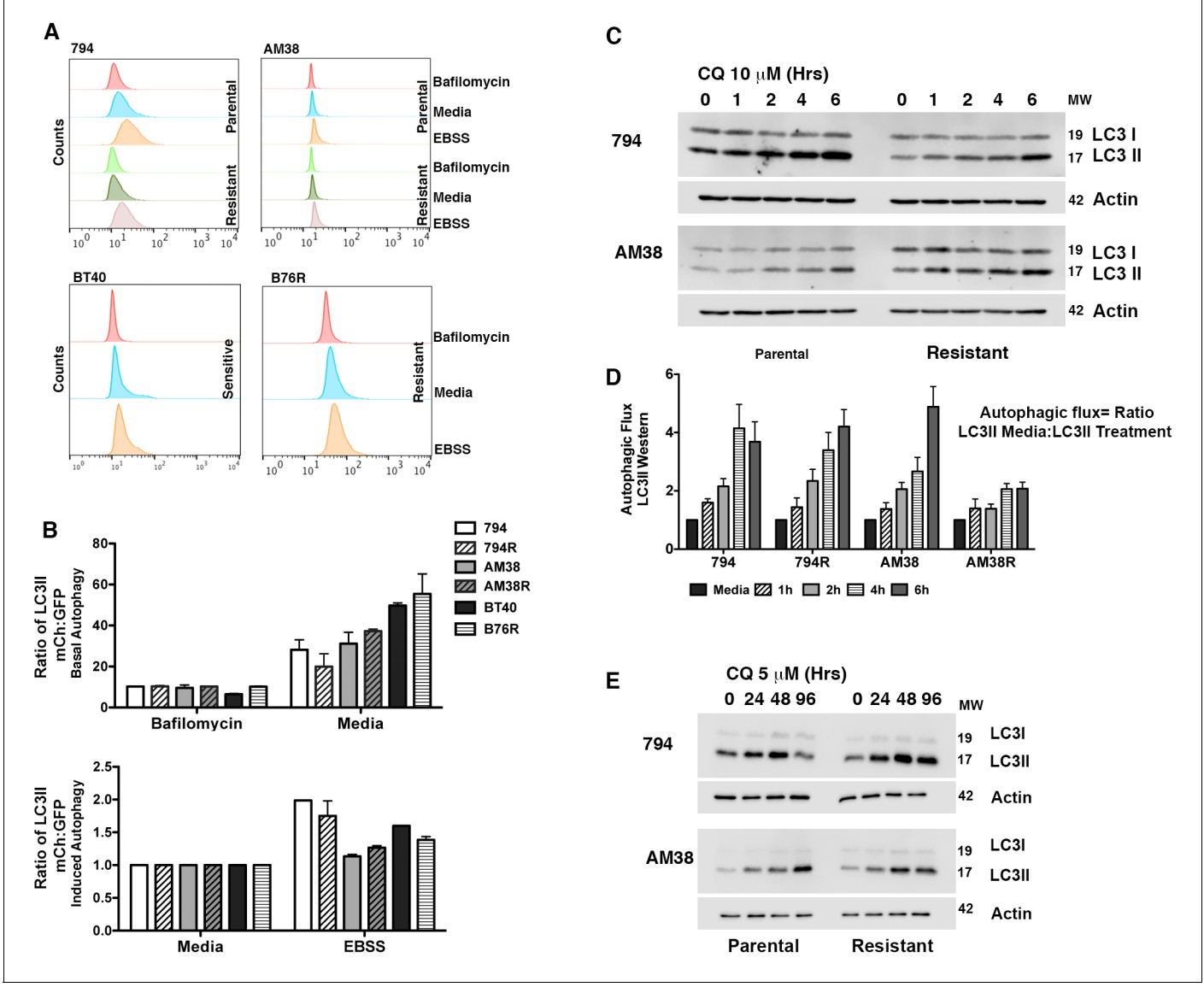

**Figure 2.** Parental and resistant BRAF[V600E] CNS tumor cells have similar levels of autophagy. (A) Representative histogram comparison of parental and resistant cell line autophagy. Cells with mCh-GFP-LC3 were exposed to either standard media or starvation EBSS media for 4 hr and analyzed by flow cytometry for the change in ratio of mCh to GFP signal as a measure of autophagic flux. (B) Quantification of basal and induced autophagy as measured in A (mean ± s.e.m, n = 3). There was no significant increase of autophagic flux in resistant over parental cell lines. (C) Representative westerns and (D) quantification of samples showing accumulation of LC3II in the presence of CQ as a measure of autophagic flux (mean ± s.e.m, n = 3). There was no significant increase of autophagic flux in resistant over parental cell lines. (E) Western blot showing inhibition of autophagy with IC50 CQ dose.

The following source data is available for figure 2:

**Source data 1.** Quantification of autophagic flux by (A) Flow cytometry) and (D) Western blotting.

showed little or no response to vemurafenib alone, but cell growth was dramatically reduced with the addition of CQ (*Figure 3B and C*), indicating a synergistic effect of combined BRAF and autophagy inhibition in both the BRAFi-sensitive cells and their resistant derivatives. Calculated combination index (CI) values for these combinations confirmed a synergistic interaction between these drugs irrespective of whether the cells had become resistant to single agent BRAFi or not (*Table 1*).

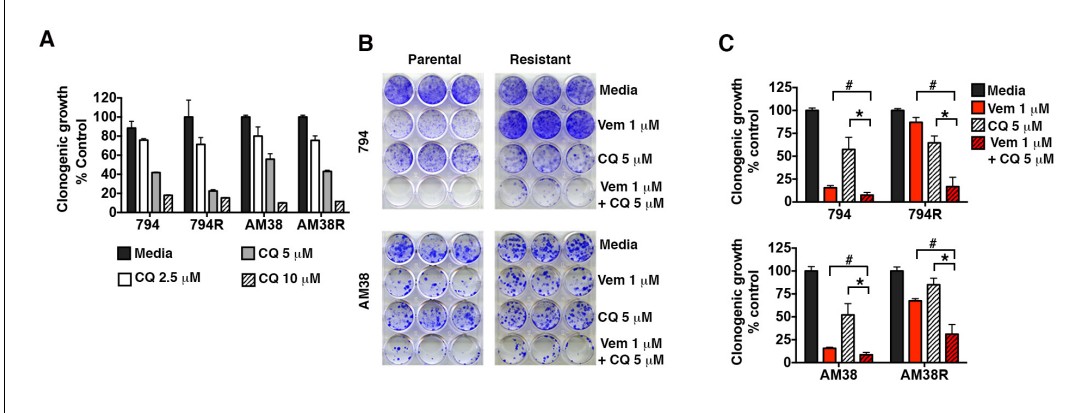

**Figure 3.** Pharmacologic inhibition of autophagy overcomes BRAFi resistance. (**A**) Long-term growth assay of parental and resistant cells in response to continuous autophagy inhibition (mean ± s.e.m, n = 3). (**B**) Representative and (**C**) quantified long-term growth of parental and resistant cells following continuous autophagy inhibition (CQ), BRAF inhibition (Vem), or combination therapy. Two way ANOVA; mean ± s.e.m, n = 3. *p<0.05, # p<0.001.

The following source data is available for figure 3:

**Source data 1.** Quantification of long-term clonogenic growth assays in 794 and AM38 parental and resistant cells treated with (**A**) increasing doses of CQ and (**B–C**) vemurafenib, CQ, or a combination of the two drugs.

Clinical evidence has suggested that cells with BRAFi resistance sometimes develop cross-resistance to other inhibitors of this pathway, specifically MEK inhibition (*Kim et al., 2013*). To test this in our resistant cells, we evaluated MEK inhibition with trametinib, which inhibits MEK1 and MEK2. 794R cells demonstrated no cross-resistance shown by a significant decrease in cell growth similar to parental 794 cells treated with trametinib. Conversely, while AM38 parental cells treated with trametinib demonstrated a significant reduction in growth, AM38R cells had a minimal decrease in growth rates (*Figure 4A*) indicating that for AM38 cells, but not 794 cells, the acquisition of resistance to vemurafenib was associated with cross-resistance to MEK inhibition. However, as with the BRAFi, when autophagy inhibition with CQ was combined with trametinib in AM38R cells, a further decrease in cell growth was demonstrated in both short and long-term assays (*Figure 4B–D*). These findings suggest that although the two cell lines had become resistant through different mechanisms, autophagy inhibition works similarly to reverse kinase inhibitor resistance in both cases.

## Genetic inhibition of autophagy overcomes BRAFi resistance in vitro

Autophagy-independent chemosensitization by CQ has been demonstrated previously (*Eng et al., 2016*; *Maycotte et al., 2012*). Therefore, genetic inhibition of autophagy was also performed to test if the responses seen were related to inhibition of the autophagic pathway or were due to another effect of CQ. Knockdowns of either ATG5 or ATG7 (two essential regulators of canonical autophagy) had a profound effect on the growth of both parental and resistant cell lines with a dramatic decrease in growth velocity. However, under conditions where there was measureable growth in the presence of RNAi, the addition of vemurafenib to resistant cells along with ATG5 or ATG7

**Table 1.** Combination index values for long-term growth assays in parental and resistant cells.

| Cell line | Vemurafenib 1 μM + CQ 5 μM |
| --- | --- |
| 794 | 0.41 |
| 794R | 0.74 |
| AM38 | 0.15 |
| AM38R | 0.85 |

R= drug induced resistance; Value > 1 antagonistic,=1 additive,<1 synergistic.

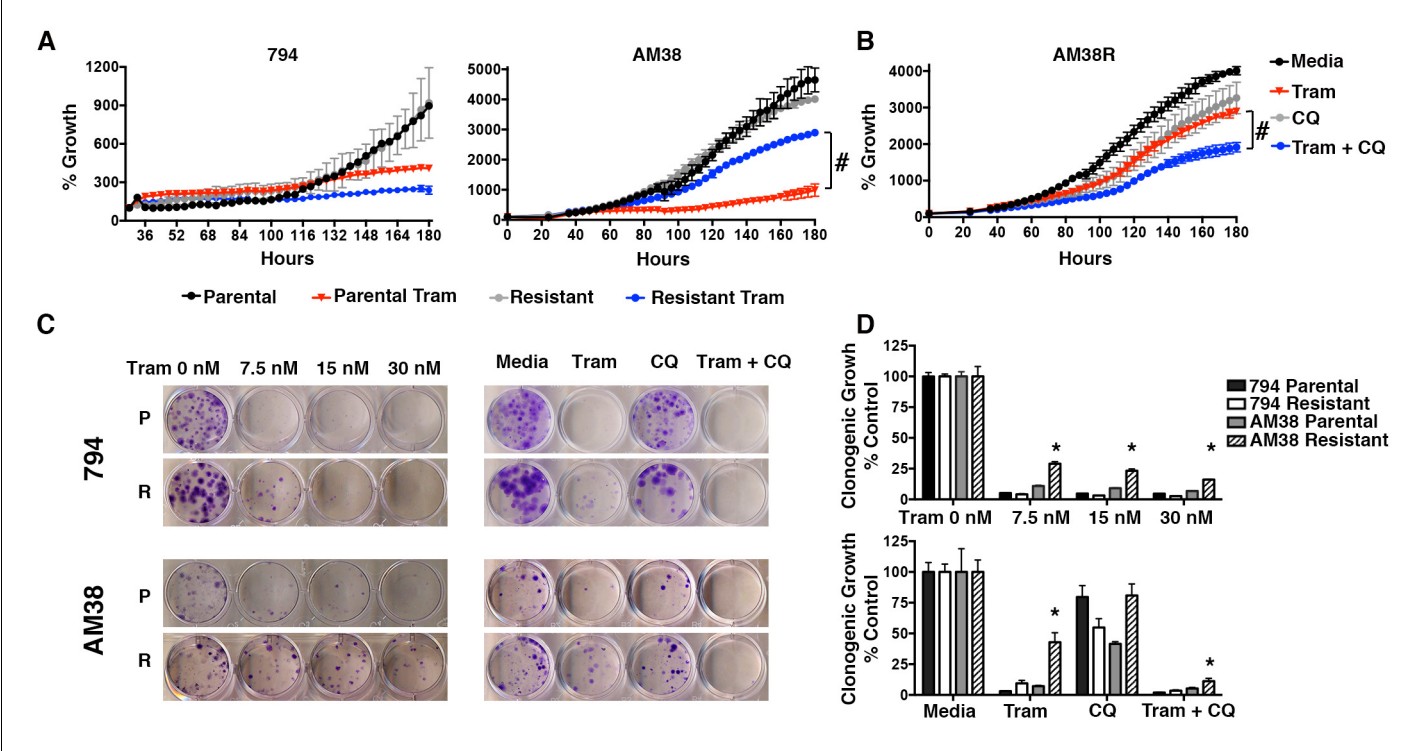

**Figure 4.** Autophagy inhibition overcomes cross-resistance to MEKi. (**A**) Percent growth over time in BRAFi resistant cell lines treated with 5 nM trametinib, a MEK inhibitor (Tram). Growth measured by continuous Incucyte monitoring (Two way ANOVA, mean ± s.e.m., n = 3, # p<0.001). (**B**) Percent growth over time in AM38R cells treated with MEK inhibition (Tram), autophagy inhibition (CQ), or combination therapy. Growth measured by continuous Incucyte monitoring (Two way ANOVA, mean ± s.e.m., n = 3, # p<0.001). (**C**) Representative and (**D**) quantified long-term growth assay of parental and BRAFi resistant cells following continuous MEK inhibition (Tram), autophagy inhibition (CQ 5 µM), or combination therapy. One way ANOVA; mean ± s.e.m, n = 3. *p<0.001.

The following source data is available for figure 4:

**Source data 1.** Quantification of (**A**) % growth over time for 794 and AM38 parental and vemurafenib resistant cells treated with trametinib alone or (**B**) cells treated with trametinib, CQ or a combination of the two drugs.

knockdown resulted in a further decrease in growth rate, under conditions where there was measurable growth in the presence of RNAi. (*Figure 5A–B*). Confirmation that ATG5 and ATG7 knockdown inhibited autophagy was shown by a decrease in the autophagy marker LC3II (*Figure 5C–D*). Combination therapy with RNAi and vemurafenib also resulted in a decreased number of viable cells compared to pharmacologic BRAF inhibition or genetic autophagy inhibition alone (*Figure 5E–F*). These data suggest that re-sensitization to the BRAF^V600E inhibitor was due to autophagy inhibition and not another effect of CQ. More importantly, both the trametinib sensitive 794R and trametinib resistant AM38R cells responded equally well to genetic interference with autophagy.

## Autophagy inhibition improves clinically-acquired BRAFi resistance ex vivo

To test if autophagy inhibition can overcome drug resistance acquired during clinical treatment, we tested slice cultures of *BRAF^V600E* and *BRAF^WT* CNS tumors from patients who in some cases had been treated with vemurafenib. This strategy allows monitoring of treatment effects in tumor cells with a supporting microenvironment ex vivo. Patient #1 was diagnosed with epithelioid glioblastoma and had rapid recurrence following standard therapy including temazolamide and radiation. At relapse, a BRAF^V600E mutation was identified, and there was a successful tumor control on vemurafenib for approximately 2 years. While on vemurafenib, a rapidly growing metastatic second recurrence developed. Following confirmation that the recurrent disease continued to harbor BRAF^V600E,

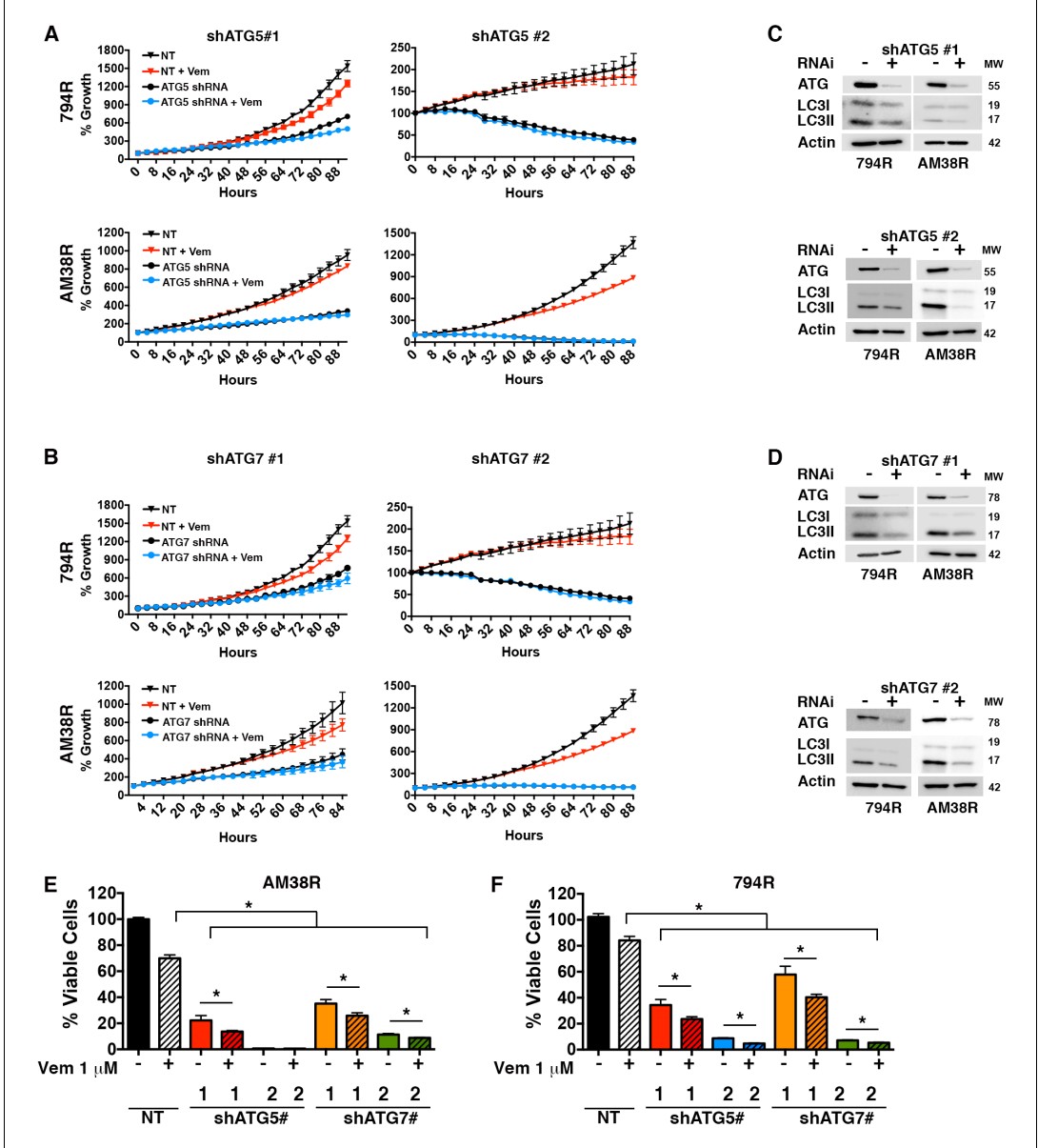

**Figure 5.** Genetic inhibition of autophagy overcomes BRAFi resistance. (A–B) Percent growth over time in resistant cell lines with control non-targeted (NT) RNAi compared to autophagy inhibition through RNAi against (A) ATG5 or (B) ATG7, required autophagy proteins. Growth measured by continuous Incucyte monitoring (mean ± s.e.m (n = 3) (C–D) Representative westerns showing effectiveness of (C) ATG5 and (D) ATG7 RNAi and resultant decrease of LC3II. (E–F) Percent viable cells, by Cell Titer-Glo (compared to control NT) following 72 hr of vemurafenib (Vem) drug therapy with and without RNAi of essential autophagy proteins ATG5 and ATG7. One way ANOVA; mean ± s.e.m (n = 3). *p<0.05.

The following source data is available for figure 5:

**Source data 1.** Incucyte timecourse and endpoint survival data.

the largest lesions were treated with focal radiation while the metastatic disease required a chemo-therapeutic approach. Tissue slices were collected at a biopsy of recurrence and evaluated for protein level changes, LDH release, and EdU incorporation following BRAFi with vemurafenib and autophagy inhibition with CQ. Western blotting demonstrated up-regulation of the pERK pathway following treatment with vemurafenib (*Figure 6A*), consistent with known resistance mechanisms that cause paradoxical activation of the pathway through other RAF isoforms or amplification of

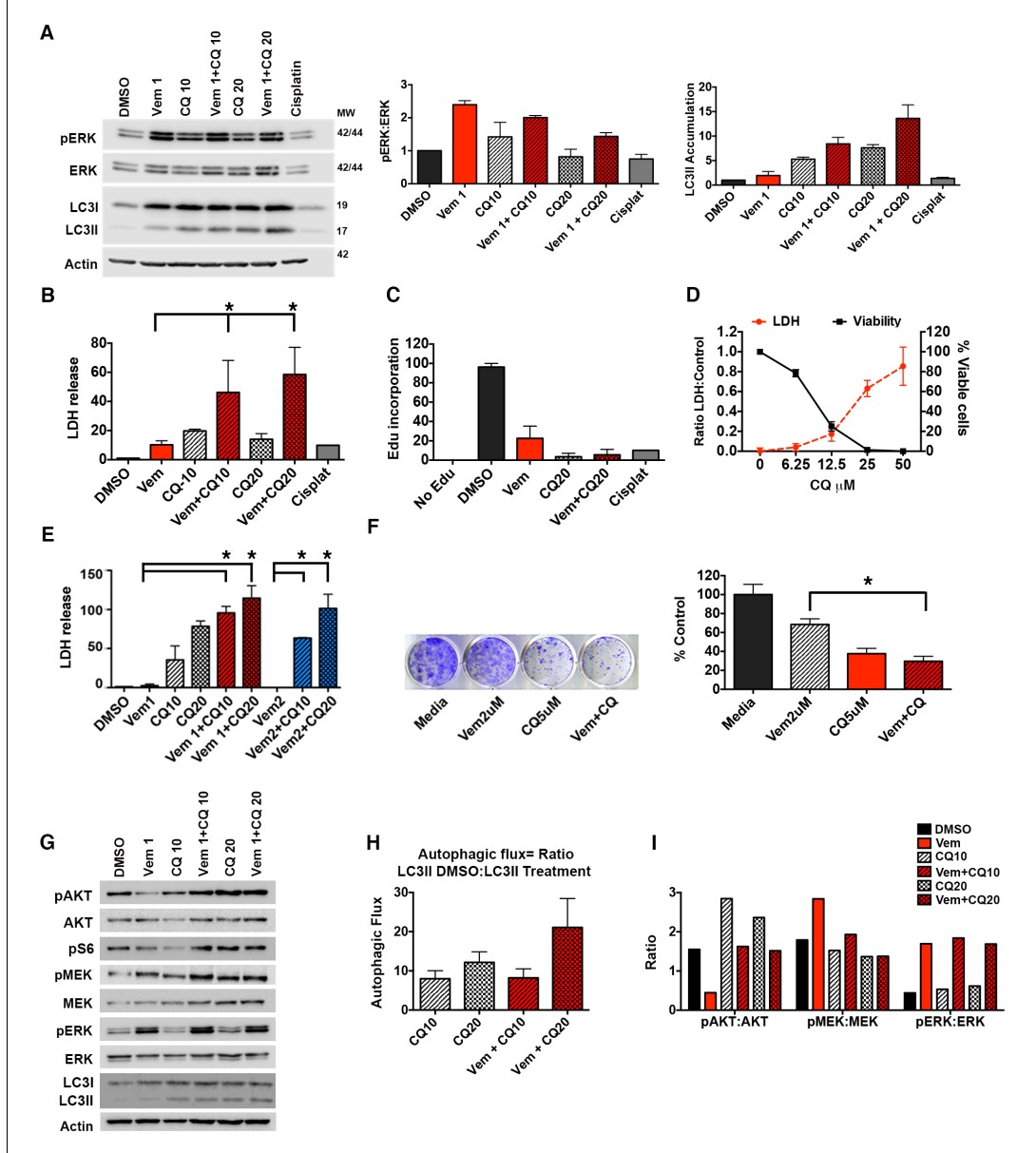

**Figure 6.** Autophagy inhibition improves clinically acquired BRAFi resistance. (**A**) Ex vivo slice culture of Patient #1 tumor showing up-regulation of pERK:ERK and inhibition of autophagic flux as indicated by LC3II accumulation by Western blot with quantification of triplicate samples, mean ± s.e.m, n = 3. (**B**) Cumulative LDH release and (**C**) EdU incorporation as a measure of cytotoxicity and decreased cell proliferation in Patient #1 treated slice culture samples; One way ANOVA; mean ± s.e.m. *p<0.05. (**D**) In vitro cell line derived from Patient #1 showing retained response to pharmacologic inhibition of autophagy with decreasing viability and contrasting increase in LDH release with increasing doses of chloroquine (CQ). (**E**) LDH release as a measure of cytotoxicity in Patient #1 cell line treated for 72 hr as indicated; vemurafenib (Vem) at 1 or 2 μM, CQ at 10 or 20 μM; Unpaired two-tailed Student's t-test; mean ± s.e.m, n = 3. (**F**) Long-term growth assay of Patient #1 cell line following autophagy inhibition (CQ), BRAFi (Vem), or combination therapy. Quantified collated data for triplicate experiments. Unpaired two-tailed Student's t-test; mean ± s.e.m, n = 3. *p<0.05. (**G**) Western blot analysis of pAKT, AKT, pS6, pMEK, MEK, pERK, ERK, LC3I and LC3II in Patient #1 slice culture samples. Actin included as loading control. (**H**) Quantification of slice culture samples showing accumulation of LC3II in the presence of CQ as a measure of autophagic flux (mean ± s.e.m, n = 3). There was no significant difference of autophagic flux between the treatment groups. (**I**) Quantified densitometry ratios of phosphorylated proteins to total proteins shown in (**G**) for AKT, MEK, and ERK.

The following source data and figure supplement are available for figure 6:

**Source data 1.** Western quantifications, LDH and survival data.

*Figure 6 continued on next page*

*Figure 6 continued*

**Figure supplement 1.** Caspase 3/7 activation occurs in the presence of BRAF and autophagy inhibition in cells with acquired BRAFi resistance.

COT (*Hatzivassiliou et al., 2010*; *Johannessen et al., 2010*; *Luke and Hodi, 2012*), but notably, also mechanistically different than when either of the BRAF$^{V600E}$ cell lines acquired resistance (*Figure 1C–D*). CQ treatment caused accumulation of LC3II within the tumor slices compared to DMSO controls, indicating successful autophagy inhibition (*Figure 6A*). Combination treatment with autophagy and BRAF inhibition resulted in significantly greater cytotoxicity than vemurafenib or CQ treatment alone as measured by LDH release (*Figure 6B*). This was associated with reduced tumor cell growth measured by EdU incorporation (*Figure 6C*).

A primary in vitro cell culture developed from Patient #1 also demonstrated sensitivity to pharmacologic inhibition of autophagy (*Figure 6D*), consistent with our data in established cell lines (*Levy et al., 2014*) and suggesting that acquisition of resistance during clinical treatment with the BRAF inhibitor did not alter autophagy-dependency. Moreover, combination therapy demonstrated significantly higher cytotoxicity (*Figure 6E*) that was associated with an increase in caspase 3/7 activity in the combination treated cells (*Figure 6—figure supplement 1*). In a long-term growth assay, combination therapy also resulted in a reduction of cell growth compared to single drug treatments (*Figure 6F*). Further pathway analysis of Patient #1 treated slice cultures was performed to evaluate the association between autophagy inhibition and the AKT/mTOR signaling pathway. Treatment with vemurafenib resulted in small decrease in pAKT and pS6 with an associated small increase in LC3II (*Figure 6A and G*) but no significant increase of autophagic flux in cells treated with vemurafenib (*Figure 6H*). This is consistent with our previously published data showing vemurafenib did not have a significant effect on the autophagic flux in parental BRAF$^{V600E}$ cell lines (*Levy et al., 2014*). As has also been previously reported (*Spears et al., 2016*), treatment with CQ resulted in increase phosphorylation of AKT. Combination therapy resulted in pAKT levels similar to DMSO control. Phospho MEK expression was also increased in all samples treated with vemurafenib, although the pMEK:MEK ratio was lower in samples treated in combination with CQ, primarily due to an increase in total MEK in these samples (*Figure 6G and I*). Importantly, in all treatments with vemurafenib, regardless of the presence of CQ, pERK was upregulated compared to DMSO control (*Figure 6A,G and I*). Together, these data suggest that patient #1's tumor acquired resistance to vemurafenib through a mechanism leading to 'paradoxical' RAF pathway activation by the drug, but that subsequent combination treatment with autophagy inhibition and vemurafenib could overcome this resistance.

## Autophagy inhibition enhances BRAFi response in multiple brain tumor types

An evaluation of additional patient samples with verified BRAF mutations allowed assessment of the effectiveness of this approach in other brain tumor types. Combination treatment of ex vivo tumor from Patient #2 with a *BRAF$^{V600E}$* positive pleomorphic xanthoastrocytoma resulted in significantly greater LDH release than vemurafenib treatment alone (*Figure 7A*). In contrast, two patients (Patients #3 and #4) with WT BRAF exhibited no increase in LDH release in any of the treatment conditions (*Figure 7B*) showing that as with our previous studies in cell lines, primary tumor samples with WT BRAF display no significant sensitivity to autophagy inhibition (*Levy et al., 2014*; *Levy and Thorburn, 2012*). In Patient #2, treatment with vemurafenib resulted in an increase in Edu incorporation while a reduction of Edu incorporation was seen in combination treated cells (*Figure 7C*). An additional primary culture from Patient #5, an adult with a *BRAF$^{V600E}$* positive glioblastoma who had not received vemurafenib therapy, demonstrated inherent BRAFi resistance. However, as with Patients #1 and #2, sensitivity to pharmacologic inhibition of autophagy was seen (*Figure 7D*). Moreover, combination therapy with both vermurafenib and CQ again demonstrated significantly higher cytotoxicity compared to single drug treatment (*Figure 7E*), and was associated with an increase in caspase 3/7 activation (*Figure 7—figure supplement 1*). Both short and long term growth assays demonstrated decreased cell growth when autophagy was inhibited with or without vemurafenib (*Figure 7F and G*). Increased tumor growth was not seen in Patient #5 vemurafenib treated cells.

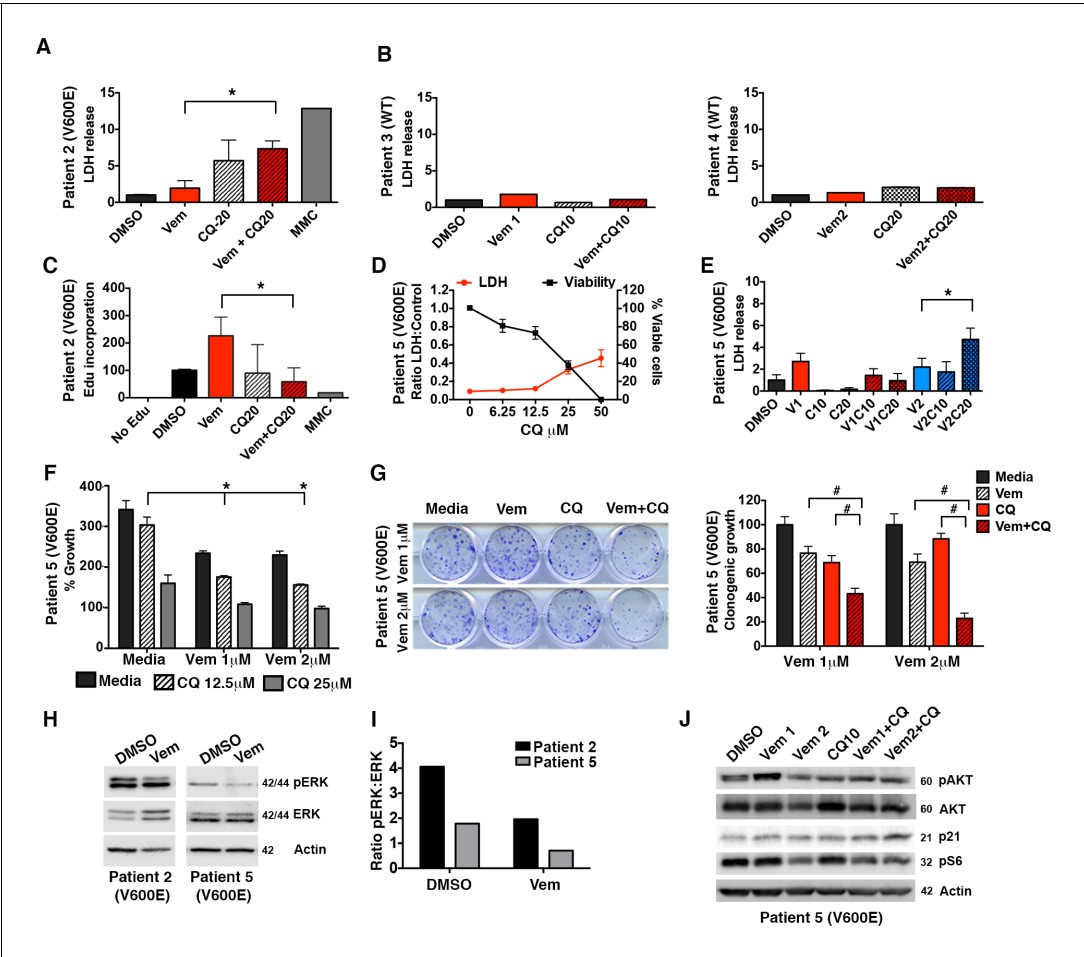

**Figure 7.** Autophagy inhibition is effective in a variety of BRAF^V600E tumor models. (**A**) Slice culture evaluation showing cytotoxicity as measured by LDH release in Patient #2 V600E mutant tumor. (**B**) No significant cytotoxicity as measured by LDH release is seen in Patients #3 and #4 with wild type (WT) BRAF tumors. (**C**) Decrease in EdU incorporation in Patient #2 V600E mutant tumor with combination BRAF (Vem) and autophagy (CQ) inhibition. (**D**) Cell line derived from Patient #5 with V600E mutant tumor showing retained response to pharmacologic inhibition of autophagy with decreasing viability and contrasting increase in LDH release with increasing doses of CQ. (**E**) LDH release in Patient #5 V600E mutant tumor cells treated with vemurafenib (Vem) at 1 or 2 µM, CQ at 10 or 20 µM autophagy inhibition (CQ), BRAFi (Vem), or combination therapy for 72 hr as indicated. Unpaired two-tailed Student's t-test; mean ± s.e.m, n = 3. *p<0.05. (**F**) Short-term (five day) growth assay demonstrating percent growth of Patient #5 cell line following autophagy inhibition (CQ), BRAFi (Vem), or combination therapy. (**G**) Representative long-term (fourteen day) clonogenic assay and quantified collated data for cells treated with combination drug therapy as indicated; Vem at 1 or 2 µM, CQ at 10 µM; Unpaired two-tailed Student's t-test; mean ± s.e.m. # p<0.001, n = 3. (**H**) Representative Western blot and (**I**) quantification demonstrating pERK response in resistant primary patient samples following BRAFi (Vem). (**J**) Western blot of PTEN downstream effectors in Patient #5 V600E mutant tumor cells, known to carry a PTEN mutation. No significant protein changes with BRAFi (Vem), autophagy inhibition (CQ), or combination therapy.

The following source data and figure supplement are available for figure 7:

**Source data 1.** Western quantifications, LDH and survival data.

**Figure supplement 1.** Caspase 3/7 activation occurs in the presence of BRAF and autophagy inhibition in cells with inherent BRAFi resistance.

The increase in Edu incorporation seen in Patient #2 was not related to paradoxical RAF pathway activation, as was seen in Patient #1 (*Figure 6A*), although there was also not a substantial reduction in pERK overall. In contrast, Patient #5' tumor cells did demonstrate a reduction in pERK signaling with exposure to vemurafenib (*Figure 7H–I*). Of note across all these patients, pERK status did not correlate with resistance to vemurafenib. Rather, inhibition of autophagy resulted in re-sensitization in all three primary patient samples. Patient #5's tumor contained additional mutations in PTEN as

**Table 2.** Mutation analysis of studied samples.

| Sample | BRAF status | Additional mutations identified |
|---|---|---|
| Patient 1 Sensitive | BRAF c. 1799T>A; p.V600E | None |
| Patient 1 Resistant | BRAF c. 1799T>A; p.V600E | None |
| Patient 5 Resistant | BRAF c. 1799T>A; p.V600E | PTEN c.74T>C; p.L25S<br>TP53 c.743G>A; p. R248Q |
| 794 | BRAF c. 1799T>A; p.V600E | None |
| 794R | BRAF c. 1799T>A; p.V600E | None |
| AM38 | BRAF c. 1799T>A; p.V600E | None |
| AM38R | BRAF c. 1799T>A; p.V600E | None |

well as a TP53 mutation (*Table 2*). Mutations in PTEN are known to confer BRAFi resistance (*Paraiso et al., 2011*), which could explain the inherent resistance found in this tumor. An evaluation of PTEN downstream effectors in Patient #5 cells found no significant effect of autophagy inhibition on pAKT, p21 or pS6 (*Figure 7J*).

## Autophagy inhibition can overcome distinct molecularly-defined BRAFi resistance mechanisms

Multiple BRAFi resistance mechanisms have been described. This includes mutations in PTEN as shown above, as well as RAS mutations and activation of receptor tyrosine kinase signaling (*Luke and Hodi, 2012*). Feedback activation of EGFR has specifically been suggested as an escape pathway in BRAF$^{V600E}$ CNS tumor cells (*Yao et al., 2015*). The above data suggest that autophagy inhibition is able to overcome BRAFi resistance in different tumor types, and that distinct mechanisms of resistance can be similarly targeted by this approach. To test this isogenic BRAF mutant cells carrying specific mutations known to confer vemurafenib resistance through distinct mechanisms were created.

RAS activation through both KRAS (*Ahronian et al., 2015*) and NRAS (*Luke and Hodi, 2012*) have been reported to result in BRAFi resistance. Using constitutively active mutants KRAS$^{G12V}$ and NRAS$^{Q61K}$, we evaluated induction of resistance and the ability to reverse this resistance by pharmacologic autophagy inhibition with CQ. 794 and AM38 cell lines expressing either KRAS$^{WT}$ or a nontarget (NT) construct retained sensitivity to both increasing doses of vemurafenib and combination therapy with CQ (*Figure 8A–D*), similar to that in parental cells (*Figure 3B*). In contrast, both KRAS$^{G12V}$ and NRAS$^{Q61K}$ cells displayed the expected resistance to increasing doses of vemurafenib alone (*Figure 8A–D*). Combination vemurafenib and CQ therapy in the RAS resistant lines resulted in a significantly increased response compared to either drug alone (*Figure 8A–D*). Calculated CI values in RAS driven resistant cells showed synergy between vemurafenib and CQ (*Table 3*). Both the KRAS$^{G12V}$ and NRAS$^{Q61K}$ cells demonstrated increased pERK activity compared to WT and NT controls, indicating the expected up-regulation of the RAF-MEK-ERK pathway (*Figure 8E*).

Because genetic autophagy inhibition is effective in reducing tumor cell growth alone and when combined with BRAFi (*Figure 5* and [*Levy et al., 2014*]), we next tested if genetic inhibition of autophagy had a similar effect when a specific, molecularly-defined resistance mechanism was modeled. ATG5 inhibition had a profound effect on cell growth in (sensitive) AM38 cells as well as (resistant) AM38R and AM38 NRAS$^{Q61K}$ cells. Due to the profound reduction in growth of cells with ATG knockdown alone (*Figure 8—figure supplement 1 A and B*), endpoint growth percentages for all shATGs utilized were compared (*Figure 8F*). Where cells were able to grow with shATG5, the addition of vemurafenib resulted in a further reduction in growth (*Figure 8F* and *Figure 8—figure supplement 1A and B*). This was also seen when a separate autophagy regulator, ATG7, was silenced (*Figure 8F* and *Figure 8—figure supplement 1C*). Moreover, efficient knockdown of ATG7 alone resulted in near immeasurable growth (*Figure 8—figure supplement 1C*) such that additional

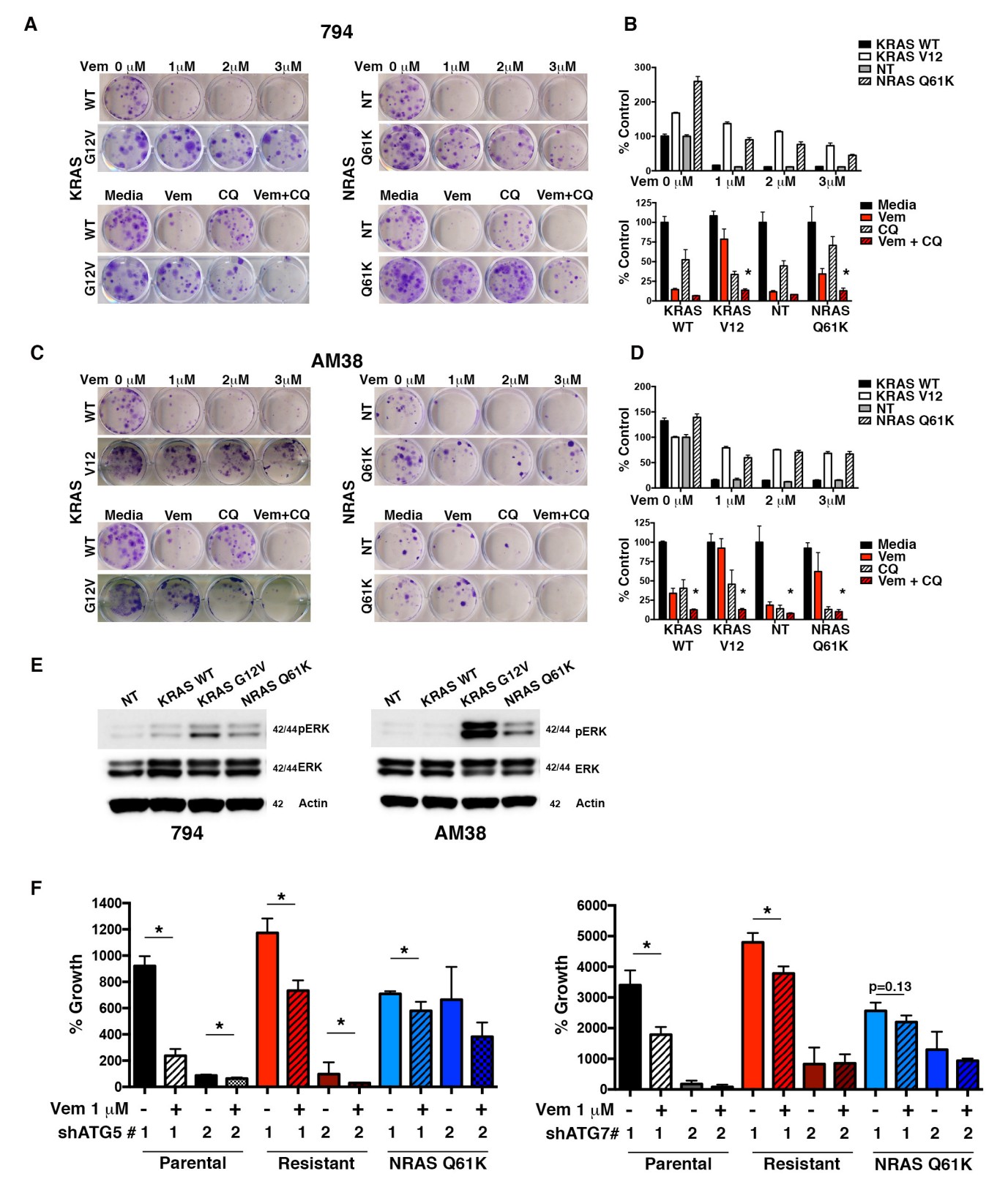

**Figure 8.** Autophagy inhibition overcomes molecularly distinct mechanisms of BRAFi resistance. (**A**) to (**D**) Representative long-term clonogenic assays (**A** and **C**) and quantified collated data (**B** and **D**) for cells treated with combination drug therapy as indicated; Vem with an increasing dose of 1, 2 or 3 μM, CQ at 5 μM; or a combination of Vem 1 μM and CQ 5 μM. Two way ANOVA; mean ± s.e.m. # p<0.001, n = 3. (**E**) Representative Westerns showing
*Figure 8 continued on next page*

*Figure 8 continued*

increased pERK expression in cells with KRAS[G12V] and NRAS[Q61K] compared to NT or KRAS[WT]. (F) Percent growth at 140 hr in AM38 (parental), AM38R (resistant) and AM38 NRAS[Q61K] (resistant) cell lines treated with autophagy inhibition through RNAi against ATG5, ATG7 or a combination of RNAi and vemurafenib. Growth measured by continuous IncuCyte monitoring. mean ± s.e.m, n = 3.

The following source data and figure supplements are available for figure 8:

**Source data 1.** Long term growth assay quantifications and incucyte timecourse data.

**Figure supplement 1.** Autophagy inhibition overcomes molecularly distinct mechanisms of BRAFi resistance.

**Figure supplement 1—source data 1.** Full image of ATG7 Western with associated actin blot for control to demonstrate shATG5 bands cut out of image.

growth inhibition with the addition of vemurafenib was difficult to detect. Representative Westerns demonstrate shRNA silencing of ATG5 and ATG7 (*Figure 8—figure supplement D and E* ).

Feedback activation of EGFR, represents another mechanistically-distinct resistance mechanism that can provide an escape pathway in BRAF[V600E] CNS tumor cells (*Yao et al., 2015*). Therefore, we also developed cell lines with EGFR overexpression (EGFRoe) and evaluated their response to autophagy inhibition. Compared to parental cells, 794 and AM38 with EGFRoe demonstrated a faster growth velocity and reduced response to vemurafenib (*Figure 9A*). When growth of EGFRoe cells was assessed for response to single drug and combination therapy, combination therapy resulted in a significantly slower growth velocity (*Figure 9B*). Representative end-point images demonstrate the reduced number of cells seen with combination therapy (*Figure 9—figure supplement 1*). Analysis of the percent of viable cells also demonstrated a significant decrease in combination therapy compared to pharmacologic BRAF or autophagy inhibition alone (*Figure 9C*). Western blotting analysis verified the increased pEGFR expression in EGFRoe cells (*Figure 9D*). It is noted that EGFR was not significantly overexpressed in our drug induced polyclonal 794R or AM38R cells. This would suggest that EGFRoe was not the main driving resistance mechanism in those cells, although as a polyclonal population the resistant cells may contain multiple resistance mechanisms and a subpopulation may contain EGFRoe.

## Autophagy inhibition decreases growth of brain tumors in patients resistant to BRAF inhibition

Based on our encouraging results reversing BRAFi resistance in different cell lines, and specifically in Patient #1's ex vivo and in vitro samples, Patient #1 was treated following recurrence on vemurafenib alone with vemurafenib at standard dosing plus 250 mg daily of CQ during focal radiation of large primary lesions. Vemurafenib was continued and the CQ dose was increased to 500 mg daily following completion of radiation to treat leptomenengeal metastatic disease sites. The dosing schedule was modeled after previously published reports using CQ for autophagy inhibition in brain tumor patients (*Briceño et al., 2003*; *Rojas-Puentes et al., 2013*; *Sotelo et al., 2006*). Consistent with the

**Table 3.** Combination index values for long-term growth assays in RAS driven resistant cells.

| Cell line | Vemurafenib 1 µM + CQ 5 µM |
|---|---|
| 794 KRAS[V12] | 0.61 |
| 794 NRAS[61K] | 0.73 |
| AM38KRAS[V12] | 0.54 |
| AM38NRAS[61K] | 0.01 |

R= drug induced resistance; Value > 1 antagonistic,=1 additive,<1 synergistic.

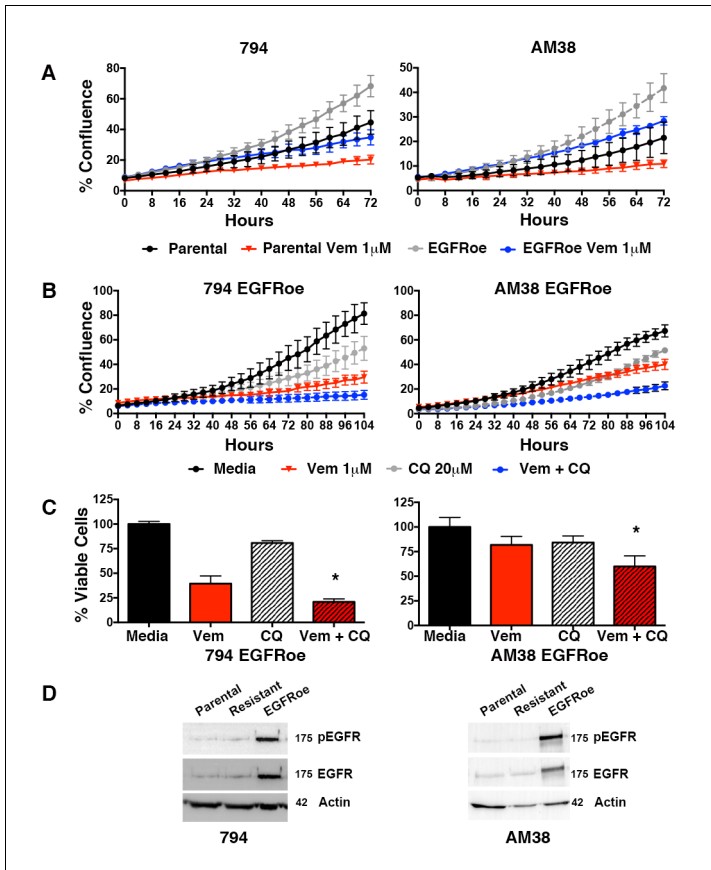

**Figure 9.** Autophagy inhibition overcomes BRAFi resistance due to escape through EGFR. (**A**) Percent confluence over time in parental and EGFR overexpression (EGFRoe) cell lines treated with BRAFi (Vem 1 μM), autophagy inhibition (CQ 20 μM) or a combination of the two. Growth measured by continuous IncuCyte monitoring (mean ± s.e.m., n = 3). (**B**) Measure of percent viable cells (compared to media control) following 4 days of BRAFi (Vem 1 μM), autophagy inhibition (CQ 20 μM) or a combination of the two. One way ANOVA; mean ± s.e.m (n = 6), *p<0.05. (**C**) Percent viable cells, by Cell Titer-Glo (compared to control NT) following four days of vemurafenib (Vem) drug therapy with and without CQ autophagy inhibition in EGFRoe resistant cells. One way ANOVA; mean ± s.e.m (n = 3). *p<0.05. (**D**) Western blot demonstrating EGFR and pEGFR overexpression in 794 and AM38 EGFRoe cells compared to parental and polyclonal resistant isogenic cells.

The following source data and figure supplement are available for figure 9:

**Source data 1.** Incucyte timecourse and endpoint survival data.

**Figure supplement 1.** Combination BRAF and autophagy inhibition results in fewer cells with EGFR overexpression.

---

ex vivo and in vitro tumor modeling, addition of CQ to the continued vemurafenib treatment resulted in clinical improvement and decreased growth of metastatic tumor sites as shown by MRI (*Figure 10A–B*). Robust LC3II accumulation in peripheral white blood cells following CQ therapy (*Figure 10C*) suggests effective autophagy inhibition at the doses of CQ used for this patient. In addition to these easily seen larger lesions, leptomeningeal enhancement throughout the subarachnoid CSF spaces improved with combination therapy as well. In stark contrast with the rapidly growing recurrent metastases, where significant growth was seen after just two weeks at the time of initial relapse, a continued favorable response to the combination therapy was maintained for 7 months. At this time, patient #1 had to stop therapy for unrelated medical issues.

This clinical response suggests that the addition of CQ to inhibit autophagy overcame vemurafenib resistance in this patient. This is consistent with our previously reported patient, who has had a

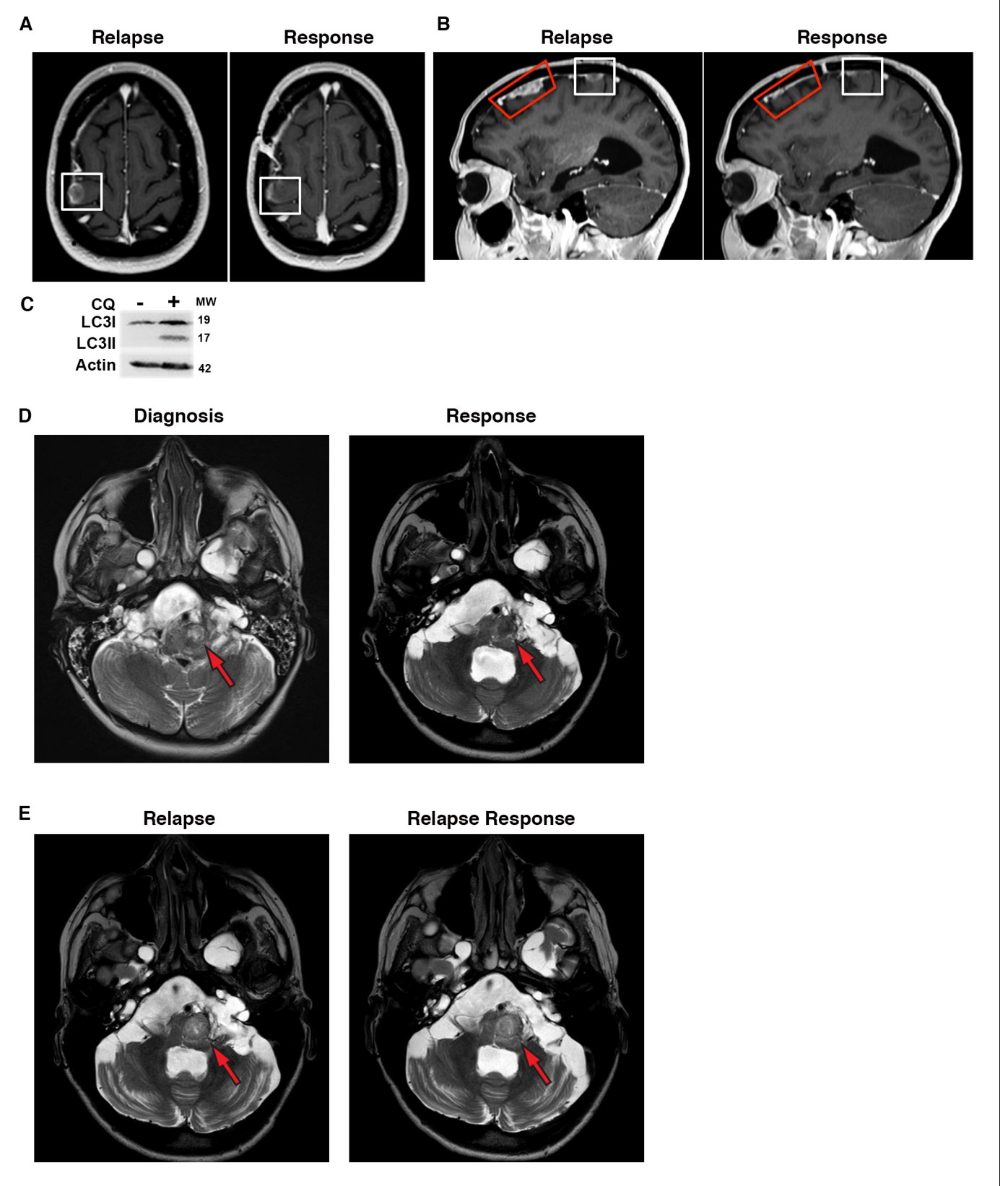

**Figure 10.** Autophagy inhibition decreases growth of metastatic glioblastoma in a patient resistant to BRAF inhibition. (**A**) Contrast-enhanced axial T1-weighted MR image demonstrates relapse of tumor over the right precentral gyrus (Relapse, white box) while receiving single drug BRAFi therapy.
*Figure 10 continued on next page*

*Figure 10 continued*

Significant response noted on the 6 month interval MRI following 6 months of combination autophagy and BRAF inhibition (Response, white box). (B) Contrast-enhanced sagittal T1-weighted MR image illustrates tumor relapse over the right frontal lobe (red and white boxes). Significant response was noted following 6 months of combination autophagy and BRAF inhibition in both radiated (Response, red box) and non-radiated (Response, white box) tumor. (C) Demonstration of clinical autophagy inhibition as measured by LC3II accumulation in white blood cells from Patient 1. (D) Axial T2-weighted MR image demonstrates a left anterolateral medullary ganglioglioma (red arrow) at diagnosis and following 1 year of vemurafenib and vinblastine therapy. (E) Axial T2-weighted MR image demonstrates a progressive left anterolateral medullary ganglioglioma (red arrow) at relapse and stable tumor following vemurafenib and CQ therapy.

durable clinical and radiographic response for over 2 ½ years with combined vemurafenib and CQ therapy following progression after 11 months of vemurafenib alone. Based on these promising clinical results, a third patient with a BRAF$^{V600E}$ brainstem ganglioglioma received CQ in addition to vemurafenib following clinical and radiologic disease progression on vemurafenib single drug therapy.

Patient #6 had a complex medical situation, including extensive multi-focal dural ectasia throughout the CNS, resulting in a severe chronic pain disorder and central hypoventilation requiring non-invasive ventilation while sleeping. This patient developed an ill-defined expansion of the brainstem with a T2 bright mass of the left anterolateral medulla (*Figure 10D*). A needle biopsy of this lesion was diagnostic for a BRAF$^{V600E}$ ganglioglioma. Initial chemotherapy with a combination of vemurafenib and vinblastine was initiated, as this had previously been shown as a successful combination in this type of tumor by our group (*Rush et al., 2013*). Patient #6 completed one year of therapy with improvement of the brainstem lesion (*Figure 10D*). It is important to note that evidence of shunt failure was present on this response scan with fourth ventricle enlargement. Due to the continued presence of tumor mass, she was maintained on vemurafenib single drug therapy for an additional six months when she presented with a worsening left facial (CN VII) palsy, increased difficulty swallowing, and balance deterioration. Associated with these worsening clinical symptoms on vemurafenib alone, progressive tumor growth was demonstrated on MRI (*Figure 10E*).

CQ (500 mg daily) was added as a second agent and she continued on vemurafenib. In contrast with her response to vemurafenib alone, and consistent with the other patients where combined therapy overcame acquired resistance to vemurafenib, within four weeks of the addition of CQ Patient #6 showed clinical improvement with improved swallowing and CN VII nerve palsy. This is a similar response to our previous brainstem BRAF$^{V600E}$ ganglioglioma patient who also showed a rapid clinical improvement with the addition of CQ therapy (*Levy et al., 2014*). This clinical improvement was maintained for two and a half months when, unfortunately, the patient developed further ventriculoperitoneal shunt failure requiring surgical intervention and her medications were discontinued to allow complete surgical healing. Within three weeks of stopping therapy, her swallowing difficulties recurred. An attempt was made to restart her medications, but complications from her swallowing difficulties resulted in an acute medical decline requiring intubation, and subsequently, she was unable to continue swallowing medications. An MRI at this time demonstrated an unchanged size of the brainstem mass (*Figure 10E*). Given the patient's inability to continue oral vemurafenib therapy and her multiple medical complications, the family elected to pursue palliative therapy only.

## Rechallenge with combination BRAF and autophagy inhibition effective in a patient with persistent BRAFi resistance

Both Patient #1 and Patient #6 required a treatment interruption to allow for healing from unrelated medical issues. In both patients this resulted in significant clinical deterioration. For Patient #1, during a six-month treatment interruption, she demonstrated progressive radiographic disease with associated clinical decline and was enrolled in a palliative care program (*Figure 11A*). Once she had resolution of her additional medical issues, she requested to restart tumor directed therapy. Resistance to BRAF inhibition has been reported to be reversible following a period of treatment interruption in melanoma (*Seghers et al., 2012*) therefore Patient #1 was restarted on single agent vemurafenib. However a short interval scan done at four weeks demonstrated rapid tumor

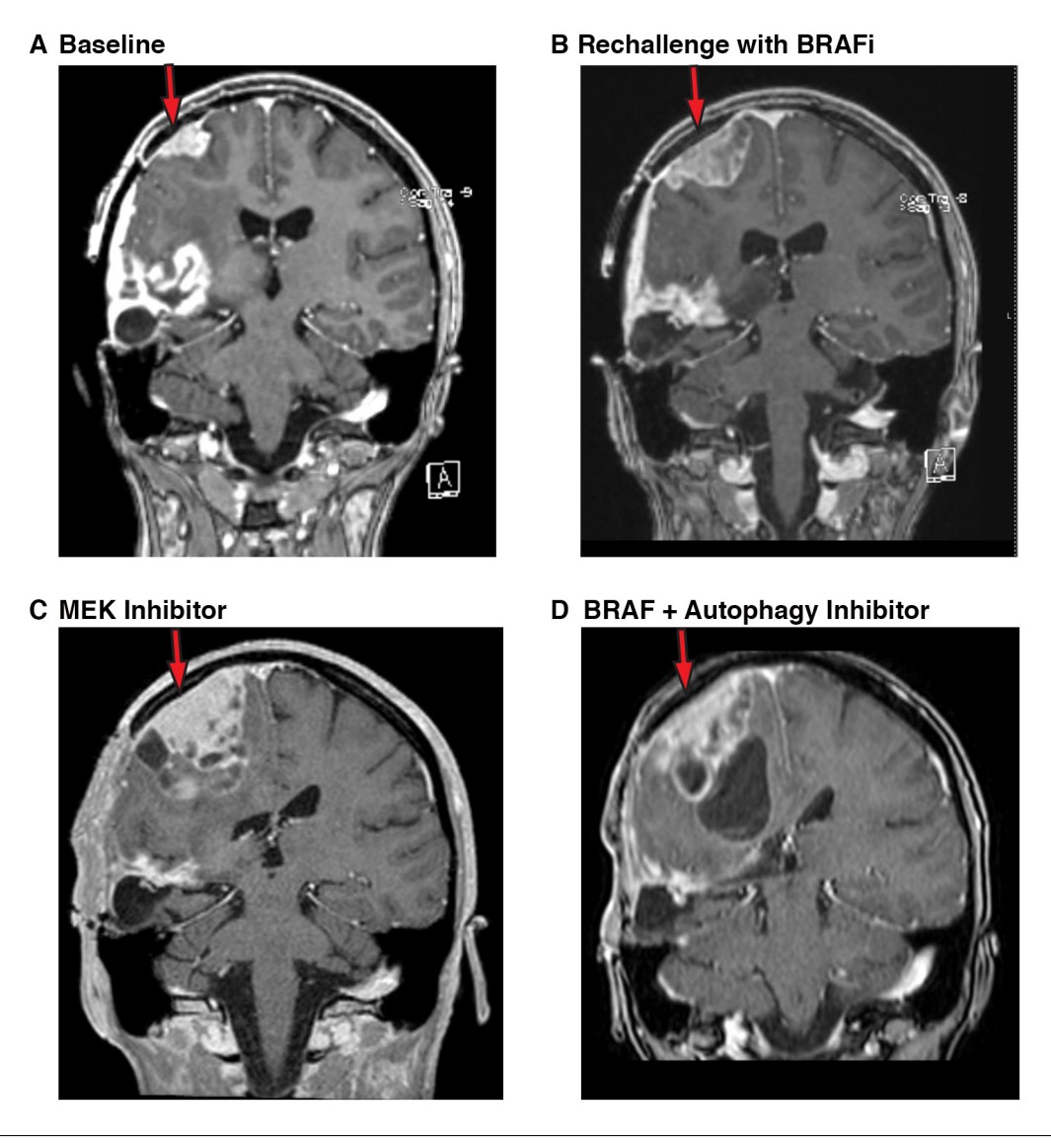

**Figure 11.** Autophagy inhibition decreases growth of metastatic glioblastoma in a patient resistant to BRAF inhibition. (**A**) Contrast-enhanced coronal T1-weighted MR image demonstrates relapse of tumor over the right precentral gyrus (red arrow) while on palliative care. (**B**) Contrast-enhanced coronal T1-weighted MR image demonstrates the progression of tumor (red arrow) following re-initiation of vemurafenib (BRAFi) single agent therapy. (**C**) Contrast-enhanced coronal T1-weighted MR image demonstrates the progression of tumor (red arrow) following trametinib (MEK inhibition) single agent therapy. (**D**) Contrast-enhanced coronal T1-weighted MR image demonstrates response of tumor with reduction in solid tumor mass (red arrow) following re-initiation of combination there with vemurafenib (BRAFi) and CQ (autophagy inhibitor).

progression on vemurafenib alone (*Figure 11B*) indicating that her tumor retained its previously acquired resistance to the BRAF inhibitor.

Combination therapy targeting both BRAF and MEK has also been reported to be well tolerated and to result in significant progression-free survival in melanoma (*Flaherty et al., 2012*; *Long et al., 2014*). It has also been reported that combination BRAF inhibition and MEK inhibition prevents MAPK pathway reactivation and an improved reduction in pERK in glioma models (*Grossauer et al., 2016*). A trial of combination vemurafenib plus trametinib was therefore initiated based on these data, but this combination was not well tolerated in this patient. Instead she was continued on single agent trametinib. After an initial stabilization lasting four weeks, Patient #1 had further clinical

decline with an evolving left hemiparesis and the development of focal seizures. Imaging repeated two months after the addition of MEK inhibition demonstrated significant tumor growth (*Figure 11C*).

In a final attempt to control tumor growth, Patient #1 was restarted on combination vemurafenib and CQ. At an evaluation four weeks after re-initiation of the combination with autophagy inhibition therapy, Patient #1 reported an overall improvement. She had regained partial use of her left side including the ability to open/close her left hand and move her left arm at the elbow. She had also regained the ability to walk unassisted for short distances where she had become wheelchair bound. An MRI at this time demonstrated a measurable reduction in her main solid tumor mass (*Figure 11D*). This patient therefore demonstrates the potential risks of cross-resistance and resistance to combination therapies targeting the same pathway. But also showed that despite a clearly highly resistant tumor, the addition of autophagy inhibition continued to have a clinical benefit.

## Discussion

Targeted therapies such as kinase inhibitors that inhibit tumor-driving mutations are expanding in importance in cancer therapy with the continued identification of mutations across tumor types and the development of many selective inhibitors of these mutant enzymes. But with this potential comes difficulties in identifying which patients to treat and how to predict and counteract the development of resistance. A recent study by Ahronian et al. evaluated resistance in colorectal cancer and highlighted many of the difficulties in combatting resistance to BRAFi (*Ahronian et al., 2015*). They found multiple resistance mechanisms across tumors and often more than one resistance mechanism in a particular patient. More importantly, they found that when cells became resistant to one combination of drugs, there was often cross-resistance to other potential combination therapies. This emphasizes the difficulty of targeting individual resistance mechanisms e.g. by adding an inhibitor of a downstream kinase such as MEK to circumvent resistance to BRAF inhibition. Tumor heterogeneity is also a significant concern with studies showing branched evolution in mutations and resistance mechanisms that can contribute to acquired resistance. Such heterogeneity can occur both temporally as well as geographically within the same patient (*Shi et al., 2014*).

The development of a therapeutic strategy that circumvents different molecularly distinct resistance mechanisms could potentially address these issues. Here we present evidence from cell lines with both experimentally acquired and different molecularly defined resistance mechanisms and from tumor samples from patients with clinically acquired and intrinsic BRAFi resistance, that pharmacological inhibition of autophagy with CQ can achieve this goal. In the cell lines where it was experimentally feasible, we could also demonstrate a similar effect with genetic inhibition of autophagy. Broad applicability is suggested because the various resistance mechanisms that could be overcome included examples that did and did not involve cross-resistance to MEK inhibition, mutational activation of both KRAS and NRAS, EGFR overexpression, paradoxical activation of the ERK pathway by vemurafenib, and a PTEN mutation. Taken together, our data suggest that regardless of the resistance mechanism to the BRAF inhibitor, autophagy inhibition was able to improve the response to BRAFi.

We have seen rapid clinical responses (in as little as six weeks) in patients with both high and low-grade tumors with acquired BRAFi resistance. These first clinical responses have also been sustained. Patient #1 demonstrated control and reduction of her metastatic tumors for greater than seven months, whereas she had previously shown MRI-measurable tumor growth in as little as two weeks. Moreover, patient #1 has experienced clinical benefit when the autophagy inhibitor was added to vemurafenib despite having demonstrated subsequent acquisition of clinical resistance to MEK inhibition after the acquisition of BRAF resistance. Also, the first patient we reported (*Levy et al., 2014*) maintained sustained tumor regression on the combination of vemurafenib and CQ for more than 2 ½ years without significant clinical complications. Although these early clinical results are encouraging, our findings are in a limited number of patients and further clinical investigation is required to verify if this strategy of combining autophagy inhibition with BRAF inhibition provides a durable and widely applicable response in BRAF$^{V600E}$ tumors.

In summary, pre-clinical and clinical experience invariably shows that tumor cells rapidly evolve ways around inhibition of mutated kinase pathways like the RAF pathway targeted here. However, based on our results, we hypothesize that by targeting an entirely different cellular process, i.e.

autophagy, upon which these same tumor cells rely, it may be feasible to overcome such resistance and thus re-establish effective tumor control. Importantly, our data suggest that this strategy can work even when different resistance mechanisms apply. This can be done using CQ, which is an approved, safe, and inexpensive drug and, perhaps, other more potent autophagy inhibitors that are under development (*Egan et al., 2015*; *Goodall et al., 2014*; *Ronan et al., 2014*). Importantly, in the context of BRAF mutant pediatric brain cancers where BRAF inhibition is already being tested, it should be feasible to quickly test this hypothesis in clinical trials.

## Materials and methods

### Study design

Experiments were designed to evaluate the hypothesis that autophagy inhibition provides a different way to circumvent BRAF inhibitor resistance in CNS tumors and might apply to multiple different mechanisms of kinase inhibitor resistance. The effect of autophagy inhibition to overcome resistance was initially evaluated in vitro in cell lines and then extended to include ex vivo studies of primary tumor samples. To ensure a complete evaluation of the effects on cell growth and death, both long and short-term growth assays were utilized as well as evaluation of LDH release and EdU incorporation as appropriate. Specificity to the autophagy pathway was evaluated with genetic inhibition studies. Final endpoints were defined prior to the start of each experiment. All in vitro experiments were completed with a minimum of three biologic replicates and where possible with triplicate technical replicates. Due to limitations in slice culture availability (only primary biopsy samples available for analysis), ex vivo tumor experiments were limited to triplicate samples from the same biopsy sample. Details on replicates and statistical analysis are indicated in the figure legends.

### Study approval

Primary patient samples were obtained from Children's Hospital Colorado and collected in accordance with local and Federal human research protection guidelines and institutional review board regulations (COMIRB 95–500). Informed consent was obtained for all specimens collected.

### Statistics

Statistical comparisons were completed using one and two way ANOVA nine and unpaired two-tailed Student's t-test (GraphPad Prism 6.0, RRID: SCR_002798) as indicated in the figure legends. A P-value of less than 0.05 was considered statistically significant. Data shown are mean ± SEM except where indicated.

### Reagents and cell lines

Vemurafenib was obtained from LC Laboratories (Woburn, MA). BT40 cells were derived from a primary patient sample and kindly provided as a gift from Dr. Peter Houghton (Nationwide Children's Hospital, Columbus, OH). The AM38 (RRID:CVCL_1070) cell line was purchased from the Japan Health Sciences Foundation Health Science Research Resources Bank (Osaka, Japan). 794, Patient #1 and Patient #5 (B76) cell lines were established from samples obtained during routine surgery at diagnosis or relapse. Cell line authentication was performed using short tandem repeat profiling and comparison with known cell line DNA profiles. Mycoplasma contamination testing was performed using a Lonza MycoAlert Mycoplasma Detection Kit (Lonza Ltd., Switzerland).

Cell lines were maintained in media supplemented with 10–20% fetal bovine serum (FBS) (Gibco, Carlsbad, CA), dependent on cell line requirements, and at 37°C in a humidified chamber of 5% $CO_2$. Cell line authentication was performed using short tandem repeat profiling and comparison to known cell line DNA profiles. Constructs utilized for inducing resistance were purchased from Addgene (RRID: SCR_002037) as follows: pBABE-Puro-KRas*G12V was a gift from Christopher Counter (Addgene plasmid # 46746), pBabe-Kras WT (Addgene plasmid # 75282) and pBabe N-Ras 61K (Addgene plasmid # 12543) was a gift from Channing Der. pBABE-puro human EGFR was constructed using SalI and SnaBI double digestion. Retrovirus particles were produced by cotransfecting GP2-293 cells (Clontech Laboratories, Mountain View, CA, USA) pBABE-puro human EGFR and Vesicular Stomatitis G protein (VSVG) using TransIT-LT1 (Mirus).

## LDH assay

For cell line assays, cells were seeded at 1000–4000 cells per well, dependent on optimal conditions per line, in 96-well plates (Corning, Corning, NY), and incubated overnight. Cells were treated with drug doses as indicated. For slice culture samples, media for each sample was collected at treatment days 0, 2, 4, 6, 8 and 10. LDH release was quantitated using the Cytoscan-LDH Cytotoxicity Assay Kit (G-Biosciences, St. Louis, MO) according to manufacturer's instructions.

## Viability assays

For short-term viability assays, cells were seeded at 1000 to 4000 cells, dependent on optimal conditions per line, in 96-well plates (Corning, Corning, NY). RNAi cells were plated 48 hr after knockdown. Cells were treated as indicated. Viable cells were measured using the Cell Titer-Glo luminescent cell viability assay (Promega, Madison, WI) following the manufacturer's protocol. All experiments were performed three times in triplicate, and the proportion of cells per treatment group was normalized to control wells.

For long-term viability assays, 750 cells were plated in 12-well plates (Corning, Corning, NY) and incubated overnight. Cells were treated as indicated. Fresh media or fresh media with drug was provided every three days until control wells had grown to approximately 80% confluence. Cells were fixed and stained using 0.4% crystal violet. Stained cells were solubilized in 33% acetic acid and absorbance was read at 540 nm. All experiments were performed three times in triplicate, and the proportion of cells was normalized to control wells.

## Synergy measurement

The combination index was calculated by the Chou-Talalay equation, which takes into account both the potency (IC50) and shape of the dose-effect curve (the M-value) (*Chou and Talalay, 1984*). Combination index values less than 1, equal to 1, and more than one indicate synergism, additive effect, and antagonism respectively.

## shRNA transfection

A pLKO system (Sigma-Aldrich, St. Louis, MO) was utilized for RNAi of autophagy related proteins. TRC numbers for shRNAs used are: ATG7 #1 (#7587), ATG7 #2 (#7584), ATG5 #1 (#151474), ATG5 #2 (151963), non-target (SHC016). Cells were transduced with lentivirus using 8 ug/mL polybrene and selected with the puromycin dose determined appropriate for each cell line. Level of targeted knockdown was determined by Western blot analysis.

## IncuCyte growth monitoring

Cells were seeded at 1000 cells per well in a 96-well plate (Costar, Corning, NY). Cells were cultured at 37° and 5% $CO_2$ and monitored using an IncuCyte Zoom (Essen BioScience, Ann Arbor, MI). Images were captured at 4 hr intervals from four separate regions per well using a 10x objective. Each experiment was done in triplicate and growth curves were created from percent confluence measurements or percent growth based on cell count per well.

## Western immunoblots

Cell lysates were harvested after treatments and time-points indicated using RIPA buffer (Sigma, St. Louis, MO) with phosphatase inhibitors (Roche, Indianapolis, IN). Membranes were blocked in TBS-Tween 5% milk and probed with primary antibodies at manufacturer recommended concentrations. Primary antibodies used were: ATG 7 (#8558S, RRID: AB_10831194), ATG5 (#12994S, RRID: AB_2630393), p44/42 MAP kinase (phosphorylated Erk1/2) (#9101S, RRID: AB_331646), p44/42 MAPK (Erk1/2) (#9102, AM: 330744); EGFR (#2232, RRID: AB_331707), pEGFR (#4407 RRID: AB_331795), p21 (#2947, RRID: AB_823586), pAKT (#4060, RRID:AB_2315049), AKT (#4685, AB_2225340), pS6 (#5364, RRID:AB_10694233) (Cell Signaling, Danvers, MA); LC3 (#NB100-2220, RRID:AB_10003146) (Novus Biologicals, Littleton, CO); Anti-$\beta$-actin (#12262, RRID:AB_2566811) Cell Signaling, Danvers, MA) was used as the protein loading control.

## MRI images

MRI images were obtained using standard protocols on a Siemens 1.5T Avanto (Munich, Germany) scanner.

## Slice culture

Slice cultures from primary tumor samples were maintained on Millicell Culture Inserts (Millipore, Billerica, MA) according to manufacturer's protocol. Briefly, three approximate 0.33 cm slices of primary tumor sample were placed onto a cell culture insert and maintained in specialized slice culture media (Neurobasal A media containing B27, glutamax, L-glutamine, HEPES and FGF). Slices were treated as indicated, and fresh media with drug, as appropriate, was changed every other day. Eight days after drug treatment EdU was added according to the Click-iT EdU Pacific Blue Flow Cytometry Assay Kit (Life Technologies, Grand Island, NY). On day 10 of treatment, tumor slices were collected for protein (Western blotting) and EdU analysis (flow cytometry). Flow data were acquired on a Gallios561 and analyzed using FlowJo. Media was collected on treatment day 0, 2, 4, 6, 8, and 10 and stored at $-80°C$ and analyzed together after day 10 for LDH as described above. Assays were performed in triplicate as tissue availability allowed.

## Flow cytometry

Cells constitutively expressing mCherry-GFP-LC3 were seeded at $2 \times 10^5$ in 60 mm plates and allowed to equilibrate overnight. Cells were exposed to either standard media, Earl's Based Salt Solution (EBSS) starvation media (Sigma, St. Louis, MO) or vemurafenib as indicated for evaluation of induced autophagy. Flow data were acquired on a Gallios561 (Beckman Coulter, RRID: SCR_008940, Fort Collins, CO) and analyzed using FlowJo V10.0.8, RRID: SCR_008520. Autophagic flux was determined by the ratio of mCherry:GFP.

## Sequencing

Library preparation was performed via the Illumina TruSight Tumor kit per the manufacturer's instructions (with minor modifications) using 110–374 ng of DNA derived from frozen or FFPE tissue. This kit amplifies selected regions of 26 cancer-related genes. Libraries were sequenced on the Illumina MiSeq platform for a targeted depth of no less than 500x for any individual amplicon. A custom-built bioinformatics pipeline utilizing GSNAP for sequence alignment and FreeBayes for variant calling was employed for data analysis. All genomic regions were verified to be covered by at least 500 sequencing reads and identified variants were manually inspected using Integrative Genomics Viewer (Broad Institute).

## Acknowledgements

Supported by an elope, Inc. StBaldrick's Foundation Scholar Award, NIH/NCI (K08CA193982), Alex's Lemonade Foundation Young Investigator Award and Department of Defense Career Development Award (W81XWH-14–0414) (JML). The Morgan Adams Foundation (NF and JML) and NIH/NCI R01CA150925 and RO1CA190170 (AT). Shared resources Cancer Center Support Grant (P30CA046934) (Flow Cytometry, Molecular Pathology, and Functional Genomics).

## Additional information

### Funding

| Funder | Grant reference number | Author |
| --- | --- | --- |
| National Institutes of Health | K08CA193982 | Jean M  Mulcahy Levy |
| U.S. Department of Defense | W81XWH-14-0414 | Jean M  Mulcahy Levy |
| St. Baldrick's Foundation | elope, Inc. St. Baldrick's Foundation Scholar Award | Jean M  Mulcahy Levy |
| Alex's Lemonade Stand Foundation for Childhood Cancer | Young Investigator Award | Jean M  Mulcahy Levy |

| Morgan Adams Foundation | Morgan Adams Research Science Award | Jean M Mulcahy Levy |
| --- | --- | --- |
| National Institutes of Health | P30CA046934 | Jean M Mulcahy Levy Andrew Thorburn |
| Olivia Caldwell Foundation | Low Grade Glioma Grant | Andrew M Donson Nicholas K Foreman |
| National Institutes of Health | R01CA150925 | Andrew Thorburn |
| National Institutes of Health | RO1CA190170 | Andrew Thorburn |

The funders had no role in study design, data collection and interpretation, or the decision to submit the work for publication.

## Author contributions

JMML, Conception and design, Acquisition of data, Analysis and interpretation of data, Drafting or revising the article; SZ, AMG, AM, KDD, DLA, BKK-D, BEF, MLG, JT, VA, AMD, DKB, DMM, TCH, MHH, ALG, Acquisition of data, Analysis and interpretation of data, Drafting or revising the article; RV, NKF, AT, Conception and design, Analysis and interpretation of data, Drafting or revising the article

## Author ORCIDs

Jean M Mulcahy Levy, http://orcid.org/0000-0002-3022-4246

## Ethics

Human subjects: Primary patient samples were obtained from Children's Hospital Colorado and collected in accordance with local and Federal human research protection guidelines and institutional review board regulations (COMIRB 95-500). Informed consent was obtained for all specimens collected.

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
