## [Decision Letter]

Thank you for submitting your article "Autophagy Inhibition Overcomes Multiple Mechanisms of Resistance to BRAF Inhibition in Brain Tumors" for consideration by *eLife*. Your article has been reviewed by two peer reviewers, and the evaluation has been overseen by a Reviewing Editor and Tony Hunter as the Senior Editor. The reviewers have opted to remain anonymous.

The reviewers have discussed the reviews with one another and the Reviewing Editor has drafted this decision to help you prepare a revised submission.

The reviewers were generally enthusiastic about your work and in particular its potential to have clinical impact. That said, they also made some important points that require further attention before we can consider your manuscript further.

Essential revisions:

The reviewers believe that some of the results relating to in vitro studies are overstated given the small sample sizes (often a single cell line). They recommend, and we agree, that shRNA studies with Atg5 and Atg7 should be conducted with two independent hairpins, and protein levels for the gene of interest should be shown.

The reviewers also noted errors in assembly of the manuscript (figures discussed out of order, legends missing or incorrect, incompletely labeled figures) that should be corrected in the revised manuscript.

---

## [Author Response]

*Essential revisions:*

*The reviewers believe that some of the results relating to in vitro studies are overstated given the small sample sizes (often a single cell line). They recommend, and we agree, that shRNA studies with Atg5 and Atg7 should be conducted with two independent hairpins, and protein levels for the gene of interest should be shown.*

We appreciate the suggestion and have completed these additional supportive experiments as recommended. New data incorporates additional hairpins against ATG5 and ATG7. Figure 5 (previously Figure 4) now includes IncuCyte growth monitoring experiments in 794R and AM38R cells with two hairpins each for ATG5 and ATG7 and is discussed in the subsection “Genetic inhibition of autophagy overcomes BRAFi resistance in vitro”. Importantly, similar results were obtained with the additional hairpins including a significant decrease in growth in 794R and AM38R cells and, in conditions where there was still some measurable growth in the presence of RNAi, the addition of vemurafenib further decreased cell growth. An evaluation of viable cells at the end of these growth experiments was also determined for each hairpin and is included in Figure 5 and also discussed in the aforementioned subsection. Western blotting demonstrates the effectiveness of these hairpins and is included in Figure 5. These data are consistent with our previous study in parental 794 and AM38 cells that also demonstrated the effectiveness of genetic inhibition of autophagy in BRAF^V600E^ brain tumor cells.^1^

Additional studies have been added to Figure 8 (previously Figure 7). All BRAF mutant cell lines investigated to date have had a consistent response to CQ (pharmacologic inhibition of autophagy) with all showing a significant improvement in response to BRAFi when treated with combination therapy. We have also demonstrated in both parental^1^ and polyclonal resistant cells (Figure 5) that genetic autophagy inhibition is effective in reducing tumor cell growth alone and when combined with BRAFi. To test the effects of genetic inhibition of autophagy in a molecularly defined mechanism of resistance, we evaluated NRAS^Q61K^ resistant cells as these cells demonstrated significant resistance to single agent vemurafenib therapy (Figure 8). In the revised manuscript, we expanded the analysis to include two hairpins each for ATG and ATG7 in AM38 parental, AM38R polyclonal, and AM38 NRAS^Q61K^ cells. Results were consistent across the three cell lines as well as across all of the hairpins. An additional Figure 8—figure supplement 1 has also been included to better evaluate the effectiveness of the addition of vemurafenib to the genetically inhibited cells. This was important as often the RNAi resulted in no growth of the cells and the combination effect upon addition of vemurafenib was therefore difficult to detect. An endpoint evaluation highlights the differences between the treatment groups that are difficult to see on the growth curves. This was particularly notable in cells with shATG7 #2, which had a very efficient knockdown and loss of protein. Western blotting analysis has been added to Figure 8—figure supplement 1 to show efficiency of RNAi knockdown.

*The reviewers also noted errors in assembly of the manuscript (figures discussed out of order, legends missing or incorrect, incompletely labeled figures) that should be corrected in the revised manuscript.*

All errors in assembly of the manuscript have been corrected. Figure 1 has been split into two figures (now Figure 1 and Figure 3) to allow the findings in Figure 2 to be discussed in proper order. All subsequent figures have been adjusted accordingly. Additionally, all figures and figure legends have been edited to ensure they are properly labeled with appropriate text.

Reference:

1) Levy JM, Thompson JC, Griesinger AM, et al. Autophagy inhibition improves chemosensitivity in BRAF(V600E) brain tumors. Cancer Discov. 2014;4(7):773-780.